# Antimalarial and neuroprotective effects of ethanolic extracts of the five-flower remedy in an experimental cerebral malaria model

Walaiporn Plirat[1,2], Prapaporn Chaniad[1,2], Arisara Phuwajaroanpong[2,3], Atthaphon Konyanee[1,2], Laddawan Lalert[4], Abdi Wira Septama[5], Chuchard Punsawad [1,2]*

1 Department of Medical Sciences, School of Medicine, Walailak University, Nakhon Si Thammarat, Thailand, 2 Center of Excellence in Tropical Pathobiology, Walailak University, Nakhon Si Thammarat, Thailand, 3 Department of Medical Technology, School of Allied Health Sciences, Walailak University, Nakhon Si Thammarat, Thailand, 4 Department of Physiology, Faculty of Medical Science, Naresuan University, Phisanulok, Thailand, 5 Research Center for Pharmaceutical Ingredient and Traditional Medicine, Cibinong Science Center, National Research and Innovation Agency (BRIN), Cibinong Science Center, West Java, Indonesia

* chuchard.pu@wu.ac.th

## Abstract

Cerebral malaria (CM), a life-threatening consequence of *Plasmodium falciparum* infection, is associated with a high fatality rate and long-term brain impairment in survivors. Despite advances in malaria treatment, effective therapies to mitigate the severe neurological consequences of CM remain limited. Consequently, novel anti-malarial drugs with different mechanisms or neuroprotective advantages are urgently required. This study aimed to explore the potential antimalarial and neuroprotective properties of the five-flower remedy (FFR), a traditional herbal formulation, in experimental cerebral malaria (ECM). Male C57BL/6 mice were induced with *Plasmodium berghei* ANKA to establish the ECM model. The ethanolic extract of FFR (600 mg/kg) was assessed both as a monotherapy and in combination with artesunate and administered for seven consecutive days starting at the onset of CM symptoms. Parasitemia levels, clinical progression, behavioral changes, and histopathological analysis of brain tissue were analyzed. The results revealed that the ethanolic extract of FFR alone improved outcomes in ECM, while its combination with artesunate significantly reduced parasitemia levels (80%), increased survival rates, reduced neurological deficits, and mitigated brain inflammation and behavioral changes. Histological analysis revealed decreased brain hemorrhage, leukocyte infiltration, and neuronal apoptosis. These promising results suggest that combining artesunate with FFR extract could be a valuable additional treatment for CM. This combination not only improves survival rates but also helps protect the brain by reducing inflammation, neurological damage, and behavioral changes. Further studies are needed to elucidate its drug interaction, mechanisms of action and potential clinical applications.

**Data availability statement:** The data associated with this study has been included in this article.

**Funding:** This work was funded by the Walailak University Plant Genetic Conservation Project under the Royal Initiation of Her Royal Highness, Princess Maha Chakri Sirindhorn (RSPG) (contract number RSPG-WU-11/2567). The funders had no role in the study design, data collection and analysis, decision to publish, or preparation of the manuscript.

**Competing interests:** The authors declare no competing interests regarding the publication of this study.

## Introduction

Malaria is an infectious disease that has long been prevalent and continues to pose a substantial public health challenge. Despite advancements in prevention and treatment, it remains a cause of morbidity and mortality [1,2]. According to the World Health Organization (WHO) report in 2024, malaria affected over 263 million individuals worldwide and resulted in > 597,000 fatalities. The projected number of malaria cases has steadily increased, with most of this growth occurring in the WHO African (89.7%) and Eastern Mediterranean (15.5%) Regions [3]. Thailand records < 10,000 malaria cases annually; however, some areas, especially forested border provinces, remain at high risk due to malaria-carrying mosquitoes. Although malaria cases occur across the country, the overall decline in infections has lowered people's natural immunity, potentially increasing the risk of severe cases [4]. *Plasmodium falciparum* is the most virulent species, responsible for the most severe and life-threatening cases of malaria [5]. Malaria initially presents with nonspecific symptoms, such as nausea and vomiting, making early diagnosis challenging. As the infection progresses, patients typically develop the classic triad of fever, chills, and profuse sweating, often accompanied by persistent headaches [6]. In some situations, the condition may advance to serious complications such as pulmonary swelling, sudden damage to the kidneys, serious anemia, yellowing of the skin shock, hemorrhaging, and cerebral issues, potentially leading to death within hours or days [7].

Although severe anemia is the most prevalent consequence, cerebral malaria (CM) is the primary cause of malaria-related deaths [8]. Patients with CM experience various symptoms, including fever, disorientation, epilepsy, and unconsciousness, which can all lead to death. Despite using typical antimalarial drugs [9], mortality rates remain high. The estimated mortality rate of CM is 20% in children and as high as 30% in adults [10], primarily due to increasing drug resistance in *Plasmodium* parasites. In addition, approximately 20% of CM survivors experience long-term neurological and behavioral complications, such as motor dysfunction, cognitive impairment, emotional dysregulation, sleep disturbances, and anxiety disorders [11,12]. The pathophysiology of CM involves two main mechanisms: Th1 mechanisms and erythrocyte sequestration. Th1 mechanisms are related to the cellular response of the immune system to the malaria parasite [13]. On the other hand, erythrocyte sequestration involves the blockage of cerebral blood vessels due to the binding of malaria parasite proteins on the surface of infected red blood cells (Parasite-encoded *P. falciparum* erythrocyte membrane protein-1) to receptor proteins on the endothelial cells of brain blood vessels. This interaction triggers inflammation within the blood vessels [14]. The immune responses to the malaria parasite can lead to inflammation and oxidative stress, causing various brain abnormalities, such as blood vessel occlusion, activation of microglial cells and astrocytes, disruptions in the structure of cerebral blood vessels, and neurotoxicity, resulting in neuronal injury and cell death [15,16]. The WHO recommends artemisinin and its derivatives as the first-line treatment for severe malaria [17].

Artesunate works by eliminating circulating ring-stage parasites, blocking their development, and preventing them from sequestering in deep organs, including the brain [18]. However, although artesunate is an effective antimalarial, it lacks specific protective effects for the brain and blood vessels, making it insufficient to fully reduce the fatality rate [19,20]. Furthermore, the widespread use of artemisinin in the treatment of malaria has led to the development of resistance in *P. falciparum* in Southeast Asia, including Thailand. The mutation of the parasite has reduced the effectiveness of the drug, causing many patients to no longer respond to standard treatment [21]. Given these limitations, developing innovative therapeutic molecules with actions distinct from currently known antimalarial medicines, or novel techniques for adjuvant therapy, requires immediate attention [22].

Botanical studies have revealed that plants and herbs are a rich source of bioactive compounds with the potential to generate novel medications for several ailments. Traditional plant-based medications have provided potent antimalarial substances, including artemisinin and quinine, which are both obtained from plant extracts [23]. Several plant-based compounds have demonstrated either direct anti-plasmodial activity or enhanced efficacy when combined with antimalarial drugs, offering promising synergy that could reduce the impact of CM. Five-flower remedy (FFR) is a Thai traditional herbal formula composed of five floral components: *Jasminum sambac*, *Mimusops elengi*, *Mesua ferrea*, *Mammea siamensis,* and *Nelumbo nucifera*. According to traditional Thai medicine, this formula is known for its rejuvenating properties, including cardioprotective properties, antipyretic effects, gastrointestinal stimulatory activity, and and the ability to alleviate symptoms of vertigo. A previous study investigated the phytochemical composition and antioxidant activity of the ethanol crude extract from FFR and its plant components. Flavonoids, terpenoids, alkaloids, and tannins were identified in the crude extracts of FFR [24]. The ethanolic extract of FFR exhibited strong free radical scavenging activity, as measured by DPPH and ABTS assays, with $IC_{50}$ values of $0.05 \pm 0.01$ mg/mL and $0.09 \pm 0.01$ mg/mL, respectively. In addition, the extract significantly inhibited nitric oxide (NO) production *in vitro*, with an $IC_{50}$ of $0.07 \pm 0.01$ mg/mL, indicating potent anti-inflammatory potential [25]. The pharmacological background of the FFR highlights the distinct bioactivities of its individual components. Among them, *Jasminum sambac* extract demonstrated the strongest antioxidant capacity, with $IC_{50}$ values of 0.82 µg/mL and 0.72 µg/mL [26], and its essential oil exhibited broad-spectrum antimicrobial activity, notably against *Klebsiella pneumoniae* [27]. Additionally, *Mammea siamensis* extract showed the highest total phenolic content, averaging $84.50 \pm 0.37$ mg GAE/g, which may contribute to its pharmacological potential [26]. An *in vitro* study assessing the antimalarial activity of individual herbal extracts against *Plasmodium falciparum* found that the ethanol extracts of *Mesua ferrea, Mammea siamensis*, *Mimusops elengi*, and the essential oil of *Jasminum sambac* exhibited a strong antimalarial activity, with $IC_{50}$ values of 4.54, 1.50, 7.39, and 0.4 µg/mL, respectively, without causing any negative effects in Vero cells [27–29]. Previous studies have demonstrated that polyherbal formulations exhibit greater antimalarial efficacy than single-herb extracts. This is due to synergistic interactions among herbal components, which enhance their potency while simultaneously reducing toxicity [30,31]. A previous study reported that the ethanolic extract of the FFR exhibited *in vitro* antimalarial activity with an $IC_{50}$ value of $2.8 \pm 0.3$ µg/mL while showing no toxicity to the Vero and HepG2 cell lines [24]. The ethanolic extract of FFR demonstrated significant in vivo antimalarial activity, with approximately 60% suppression of parasitemia, and showed no signs of toxicity in a murine model (unpublished data). However, no studies have specifically investigated the neuroprotective and antimalarial effects of FFR against CM. Given its traditional medicinal use and promising biological properties, this herbal formula may represent a potential treatment option for managing CM and preventing its associated neurological complications.

Therefore, this study aimed to investigate the neuroprotective effects of the FFR in an experimental CM (ECM) model. The objective is to determine whether this traditional herbal formulation can prevent or reduce the severity of neurological abnormalities associated with CM. The findings will serve as fundamental data for future clinical trials, ultimately supporting the use of Thai herbal medicine as an alternative therapy for the prevention and treatment of CM.



## Materials and methods

### Plant materials

The plant samples used in this study include *Nelumbo nucifera*, *Jasminum sambac*, *Mimusops elengi*, *Mesua ferrea*, and *Mammea siamensis.* The specimens were obtained as dried herbal products from a traditional Thai pharmacy store in Muang District, Nakhon Si Thammarat Province, Southern Thailand. Each specimen was taxonomically identified and scientifically named by an expert (a specialized botanist connected with Walailak University's School of Pharmacy) to verify accuracy and compliance with botanical classification criteria. Voucher specimens were identified and placed in the Department of Medical Sciences, School of Medicine, Walailak University, Thailand. Table 1 shows the five plants.

### Plant preparation and extraction

The plant samples were thoroughly washed, air-dried, and subsequently oven-dried at 50°C. The dried samples were coarsely ground using a grinder (Taizhou Jincheng Pharmaceutical Machinery Co., Ltd., Model; SF, Jiangsu, China), weighed, and stored in airtight containers. To prepare the ethanolic extract of the FFR, 12 g of each plant species were combined, resulting in a total weight of 60 g. The mixture was then coarsely ground, and these portions were extracted using 95% ethanol. A 60-g sample was placed in a maceration flask, followed by the addition of 600 mL of ethanol. The extraction process took 3 days, and the extract was then filtered through Whatman No.1 filter paper (Whatman, Buckinghamshire, England). The plant residue was re-extracted twice more using the same procedure. The solvent was then evaporated using a rotary vacuum evaporator at 50°C (Rotavapor, Buchi, China), followed by further evaporation in a water bath to obtain the ethanol extract. The ethanol extracts were weighed to determine the total extract yield. The extract was stored in airtight containers at 4–8°C until further analysis. For optimal stability, the crude extract should be used within 2 months to preserve its bioactivity.

### Ethics approval

The Walailak University Institutional Animal Care and Use Committee granted ethical permission for the animal-based investigations before the experiments began (approval number: WU-ACUC-67009). Research and animal care professionals received extensive training in the proper handling and usage of laboratory animals. All experimental protocols properly followed applicable ethical rules and regulations governing animal welfare, including full conformity with the Animal Research: Reporting of In Vivo Experiments requirements. Surgical procedures were conducted under controlled isoflurane anesthesia, and every effort was made to minimize animal distress and suffering throughout the trial. In addition, continual monitoring was conducted to ensure the well-being of the animals during the experiment.

### Experimental model and malaria parasite

Male C57BL/6 mice, aged 6–8 weeks and weighing approximately 20–25 g, were purchased from Nomura Siam International Co., Ltd. The mice were housed in a controlled laboratory setting with a temperature range of 22–25°C,

**Table 1. Plants components in five-flower remedy [24].**

| Plant ingredients | Common name | Part of used | Family | Voucher number |
|---|---|---|---|---|
| *Jasminum sambac* Ait | Arabian jasmine | Flowers | Oleaceae | SMD187007002 |
| *Mimusops elengi* L. | Spanish cherry | Flowers | Sapotaceae | SMD249006002 |
| *Mesua ferrea* L. | Ceylon ironwood | Flowers | Calophyllaceae | SMD122007001 |
| *Nelumbo nucifera* | Sacred lotus | Flowers | Nelumbonaceae | SMD181001001 |
| *Mammea siamensis* | Negkassar | Flowers | Calophyllaceae | SMD122006002 |

relative humidity of 50–60%, and a 12-hour light/dark cycle. They were provided with standard pelleted food ad libitum, sterilized bedding, and clean drinking water throughout the study. All procedures followed the National Research Council's ethical guidelines for animal research, ensuring proper nutrition, minimal stress, and disease prevention. Researchers carefully monitored and documented housing conditions, feeding, handling, and overall animal welfare daily throughout the study [32].

*Plasmodium berghei* ANKA strain, a rodent malaria parasite, was obtained from the Biodefense and Emerging Infections Research Resources Repository under the National Institute of Allergy and Infectious Diseases and National Institutes of Health. Red blood cells infected with *P. berghei* were administered intraperitoneally into donor mice in 0.1 mL of phosphate-buffered saline solution. Once a parasitemia level of 20–30% was reached, blood was aseptically collected via cardiac puncture into heparinized tubes for subsequent injection into experimental mice. This method is designed to minimize potential distress and ensure the well-being of the animals throughout the procedure. The efficiency of anesthetic induction in mice was carefully evaluated by measuring the toe-pinch response to validate the lack of pain perception. Once deep anesthesia was achieved, blood samples were obtained through cardiac puncture using a sterile technique and collected in heparinized tubes to prevent coagulation. The obtained blood was properly diluted with 0.9% physiological saline and subsequently used for injection into the experimental mice. To ensure the study's integrity and animal welfare standards, all processes were carried out strictly in compliance with ethical and scientific criteria. All experimental animals were monitored daily for clinical signs indicative of cerebral malaria and general physiological distress. These signs included changes in behavior, posture, motor activity, responsiveness, and body temperature. Humane endpoints were rigorously observed in accordance with "Guidelines for Humane Endpoints" and the approved institutional animal care protocol. Mice exhibiting signs of severe disease—such as prolonged seizures, coma, complete immobility, absence of body extension reflexes, unresponsiveness to external stimuli, irregular breathing, or hypothermia—were promptly euthanized to prevent unnecessary pain and suffering. Euthanasia was carried out using deep anesthesia induced by isoflurane inhalation, followed by cervical dislocation, as stipulated by institutional guidelines. The animals was monitored for a period of up to 13 days following *Plasmodium berghei* ANKA infection. Animals that died spontaneously before humane intervention could be applied were also documented. Throughout the study, all efforts were made to minimize animal suffering and reduce the number of animals used without compromising the scientific integrity of the investigation.

## Dosing and grouping of CM model

In this study, mice were randomly categorized into five groups (10 per group) (Table 2). Group 1 (Control uninfected group) received a single intraperitoneal (IP) injection of a vehicle solution on Day 0. From Day 6–12, they were administered 7% Tween-80 and 3% ethanol orally, once daily. Group 2 (Negative control (*Pb*A-infected mice)) received a single IP injection of $1 \times 10^7$ parasites (0.1 mL) on Day 0. From Day 6–12, they were administered 7% Tween-80 and 3% ethanol orally, once daily. Group 3 (Positive control (Artesunate, Art)) received a single IP injection of $1 \times 10^7$ parasites (0.1 mL) on Day 0. From Day 6–12, they were treated with artesunate (6 mg/kg body weight) orally, once daily. Group 4 (FFR extract

**Table 2. Group classifications and doses used in the experimental cerebral malaria model.**

| Group | Drug/Extract | Dose of treatment (mg/kg) |
|---|---|---|
| Control uninfected | 7% Tween-80 and 3% ethanol | – |
| *Pb*A-infected group | 7% Tween-80 and 3% ethanol | – |
| Artesunate | Artesunate | 6 |
| Five-flower remedy | Five-flower remedy | 600 |
| Artesunate combined five-flower remedy | Artesunate and Five-flower remedy | 6 600 |

alone) received a single intraperitoneal injection of 1 × 10⁷ parasites (0.1 mL) on Day 0. From Day 6–12, they were administered FFR extract (600 mg/kg body weight) orally, once daily. Group 5 (a combination of artesunate and FFR extract (Art + FFR)) received a single IP injection of 1 × 10⁷ parasites (0.1 mL) on Day 0. From Day 6–12, they were administered artesunate (6 mg/kg body weight) along with FFR extract (600 mg/kg body weight) orally, once daily.

On day 0, male C57BL/6 mice received an IP injection of 0.1 mL of infected blood, which included 1 × 10⁷ *Pb*A parasite-infected red blood cells (pRBCs). Six days following infection, the mice were randomly categorized into five groups, as previously described. Each group was administered a daily dose of the crude extract for 7 days. To imitate the usual route of administration, the therapy was given via oral gavage. The evolution of the experimental model was tracked by measuring parasitemia levels, clinical symptoms, and body weight changes. From day 4–12 after infection, all experimental animals were examined for parasitemia and clinical symptoms of CM using parasitemia analysis and the Rapid Murine Coma and Behavior Scale (RMCBS). On days 12–13 after infection, the Novel Object Recognition (NOR) test was used to examine behavioral factors such as movement patterns and memory retention. Following behavioral testing, mice were anesthetized with 2% isoflurane (Piramal Pharma, PA, USA) via inhalation utilizing a rodent anesthetic system before being euthanized through a heart puncture. Following euthanasia, brain tissue was obtained to investigate histopathological changes and gene expression.

## Humane euthanasia procedure for animals

At the conclusion of the experiments, mice were humanely euthanized to minimize suffering and anguish. The animals were initially placed in an anesthetic induction chamber and given deep isoflurane anesthesia (Piramal Critical Care, PA, USA), which caused unconsciousness and efficiently eased any pain in accordance with accepted ethical guidelines [34]. To guarantee total euthanasia and in accordance with the American Veterinary Medical Association recommendations for animal euthanasia (2020 edition), a secondary cervical dislocation procedure was used. This method was used to confirm the death of the animals and ensure a swift, humane procedure. Throughout the process, every effort was made to adhere to the highest standards of animal welfare, minimizing distress and ensuring the ethical integrity of the study.

## Parasitemia analysis

Parasitemia in red blood cells was analyzed by collecting blood from the tails of the mice using the procedure described in our previous reports [33–35]. First, the tip of the tail was cleaned with 70% alcohol and dried. A sterile surgical scissor was used to remove approximately 0.1 mm of the tail tip. A drop of blood was applied to a glass slide to create a thin smear. The smear was stained using the Wright–Giemsa stain and viewed under a light microscope (Olympus CX31, Tokyo, Japan) at 100X oil immersion magnification to determine infection levels and the antimalarial activity of the extract. The formula for calculating the amount of infected red blood cells and percent parasitemia suppression is as follows:

$$\% \text{ Parasitemia} = \frac{\text{Number} \sim \text{of} \sim \text{infected} \sim \text{RBCs}}{\text{Total} \sim \text{RBCs} \sim \text{counted}} \text{X100}$$

$$\% \text{Parasite suppression} = \frac{\text{Mean parasitemia in the untreated group} \, - \, \text{Mean Parasitemia in the treated group}}{\text{Mean parasitemia in the untreated group}} \text{X100}$$

## RMCBS

On the fourth day after infection, behavioral clinical symptoms and illness severity were assessed quantitatively using the RMCBS grading methodology. The RMCBS procedure was used to monitor behavioral changes in experimental mice. The severity of behavioral alterations correlates with the degree of neurological impairment, making this protocol a reliable indicator of CM [36]. The RMCBS protocol evaluates 10 behavioral parameters, including gait, balance, motor movement,

body position, limb strength, touch escape, pinna reflex, toe pinch, aggression, and grooming. During the assessment, each mouse was video-recorded for 3 min. Movement-related behaviors were analyzed during the first 90 s, while limb strength and balance were assessed in the last 90 s. The RMCBS test were conducted by an independent observer who was blinded to the treatment group. Specifically, during data collection and scoring, the investigator was not informed of the treatment conditions assigned to each animal. Each parameter was evaluated on a scale of 0–2, with 0 indicating the most severe impairment and 2 indicating the least impairment. The total RMCBS score ranges from 0 to 20. The scores for each mouse were recorded throughout the study for each experimental group.

### NOR test

The NOR test is a behavioral technique used to assess recognition-based learning and memory in experimental mice [37]. The test is conducted in an open, square-shaped box measuring 33 × 33 × 20 cm³ and consist of three sessions: habituation, training, and testing (Fig 1). During the habituation session (Day 1), each mouse is placed in an empty box for 5 min to familiarize itself with the environment. In the training session, the mouse is allowed to explore two distinct objects (Object A and A') for 5 min. The time spent exploring each object is recorded. Exploration is defined as the nose touching the object, nose within 2 cm of the object, front paw interaction with the object while maintaining nose contact (climbing or sitting on the object is not counted). During the testing session (24 h after the training session), the mouse is placed in the same box with one familiar object (Object A) in its original position and a new object (Object B) replacing Object A. The mouse is allowed to explore for 5 min, and the time spent interacting with each object is recorded. All objects and the test chamber are cleaned with 70% alcohol before each session to eliminate olfactory cues. The NOR test was conducted by an independent observer who was blinded to the treatment group to minimize potential observer bias. During both the data collection and scoring phases, the investigator was uninformed of the particular treatment conditions assigned to each animal. For data analysis, recognition memory is assessed using the discrimination index (DI), which is calculated using the following formula [38]:

$$DI = \frac{Time \sim exploring \sim the \sim novel \sim object - Time \sim exploring \sim the \sim familiar}{Total \sim exploration \sim time}$$

### Brain histopathological examination

Histopathological analysis was conducted following standardized laboratory protocols as described in previous studies [39,40]. On day 13 post-infection, mice were anesthetized with 2% isoflurane (Piramal Pharma, PA, USA) via inhalation

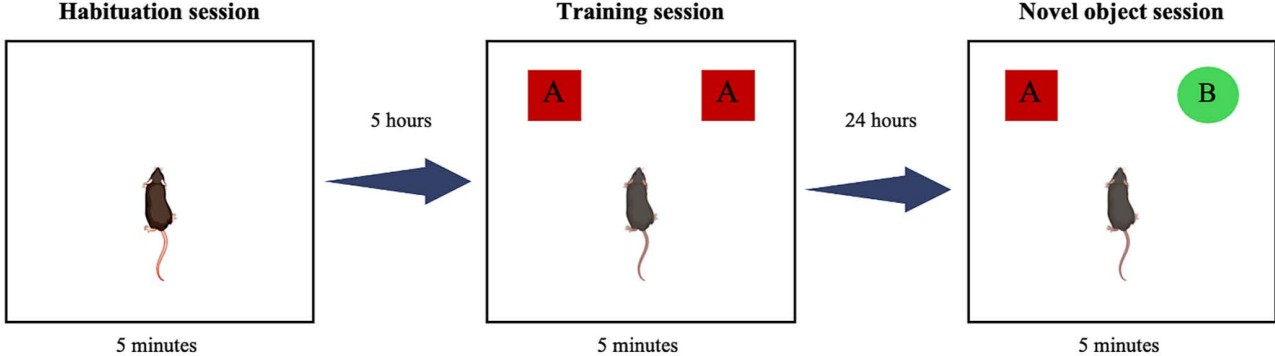

**Fig 1. Graphical overview of the novel object recognition test.**

using a rodent anesthetic system before being euthanized through heart puncture under aseptic conditions. This study focuses on the morphological changes in the cerebral cortex and hippocampus when brain tissue is processed with ethanol at various concentrations before being embedded in paraffin. The paraffin-embedded brain tissue blocks were sectioned at a thickness of 5 mm and placed on glass slides for Hematoxylin and Eosin staining to examine brain morphology. The tissue slides were deparaffinized in xylene twice, each for 1 min. The slides were rehydrated in 100% ethanol twice (1 min each) and then in 95% ethanol twice (1 min each). After cleaning with tap water, the slides were wiped to eliminate extra moisture. The slides were stained with Hematoxylin for 15 min, rinsed in tap water, and dipped 1–2 times in acid alcohol to remove excess stain. The slides were blotted to remove excess water and stained with Eosin for 3 min. Subsequently, the slides were dehydrated through 95% and 100% ethanol, followed by clearing in xylene. Finally, the slides were mounted using Permount and covered with a cover slip. The prepared slides were then examined under a light microscope to assess morphological changes in the cerebral cortex and hippocampus. Histopathological examination of the brain tissue slides was performed by an independent observer who was blinded to the treatment group. This blinding was implemented to minimize bias and ensure the objectivity of the microscopic evaluations. The observer was not provided with any information regarding the experimental groups during the analysis process.

## Analysis of gene expression of pro-inflammatory cytokines and neurotrophins factor in the brain

On day 13 post-infection (D13), brain tissue was collected for RNA isolation using GENEzol™ reagent (Catalog No. GZR100, Geneaid, Taiwan) in accordance with the instructions of the manufacturer. After homogenization, phase separation was accomplished by adding chloroform, rapidly shaking the mixture, and incubating at room temperature for 5 min. The samples were centrifuged at $12,000 \times g$ for 15 min at 4°C. The resultant aqueous phase was mixed with an equivalent volume of 70% ethanol and transferred onto an RNeasy Mini column (Qiagen, Valencia, CA, USA) for further processing, following the instructions of the manufacturer. RNA yield and purity were assessed using a Nanodrop, while RNA quality was evaluated using an Agilent 2100 Bioanalyzer (Agilent Technologies, Palo Alto, CA, USA). Only samples with an optical density between 1.8 and 2.0 were included in the experiment. The purified RNA was subsequently reverse-transcribed into complementary DNA using the iScript™ Reverse Transcription Supermix for Quantitative real-time PCR (qRT-PCR) (Catalog No. 1708840, Bio-Rad, USA).

qRT-PCR was performed using a QuantStudio™ 3 Real-Time PCR System with 5X HOT FIREPol® EvaGreen® qPCR Mix Plus (ROX). The thermocycling procedure was as follows: initial denaturation at 95°C for 15 min, followed by 34 cycles of 95°C for 30 s, 62°C for 30 s, and a final extension at 72°C for 15 min. Each experimental condition was tested in at least two independent trials. Gene levels were normalized to GAPDH, and relative expression was calculated using the ΔΔCt method. The primer sequences used in this study are listed in Table 3.

**Table 3. List of primers used for the five-flower remedy treatment in experimental cerebral malaria.**

| Gene | Primer | Sequence 5′-3′ |
|---|---|---|
| TNF-α | Forward | 5′-CTCCCTTTGCAGAACTCAGG-3′ |
| | Reward | 5′-AGCCCCCAGTCTGTATCCTT-3′ |
| IL-1β | Forward | 5′-CTAAAGTATGGGCTGGACTG-3′ |
| | Reward | 5′-GGCTCTCTTTGAACAGAATG-3′ |
| BDNF | Forward | 5′-TGGCCCTGCGGAGGCTAAGT-3′ |
| | Reward | 5′-AGGGTGCTTCCGAGCCTTCCT-3′ |
| Trk B | Forward | 5′-TGGACCACGCCAACTGACAT-3′ |
| | Reward | 5′-GAATGTCTCGCCAACTTGAG-3′ |
| GAPDH | Forward | 5′-ACACATTGGGGGTAGGAACA-3′ |
| | Reward | 5′-AACTTTGGCATTGTGGAAGG-3′ |

## Statistical data analysis

All data in this study were presented as mean±SEM. Statistical analysis was performed using SPSS version 29. All variables for each parameter were assessed for normality using the Kolmogorov–Smirnov test. Comparisons of mean values between experimental groups, including parasitemia percentage, suppression percentage, RMCBS score, body weight, mRNA levels, and behavioral assessments, were conducted using a one-way analysis of variance followed by Bonferroni's post hoc test. In this study, a confidence level of 95% was adopted, and $p$-values less than 0.05 were considered statistically significant.

## Results

### Neurological manifestations and antimalarial efficacy in ECM

By day 6 post-infection, approximately 80% of $Pb$A-infected C57BL/6 mice exhibited pronounced neurological symptoms, including reduced physical activity, muscle weakness, abnormal gait patterns, impaired righting reflexes, and a lack of aggression. By day 13 post-infection, the $Pb$A-infected group demonstrated CM symptoms, such as ataxic gait, motor dysfunction, signs of fur deterioration, loss of right reflexes, seizures, and convulsions. In contrast, mice that received artesunate combined with the FFR extract did not develop CM-related neurological symptoms and survived until the end of the study (Fig 2). Parasitemia levels were monitored in all treatment groups from day 4–13 post-infection. The $Pb$A-infected untreated group exhibited a progressive increase in parasitemia between days 5 and 13 post-infection, reflecting disease progression. By day 13 post-infection, monotherapy with the FFR extract (600 mg/kg body weight) resulted in a significant parasitemia suppression of 64.22±2.31% compared to the $Pb$A-infected untreated group ($p < 0.05$) (Table 4). Furthermore, adjunctive therapy with artesunate and FFR extract significantly reduced parasitemia levels between days 7 and 13 post-infection, with an average suppression of 77.94±1.86%, compared to the $Pb$A-infected untreated group ($p < 0.05$) (Table 4).

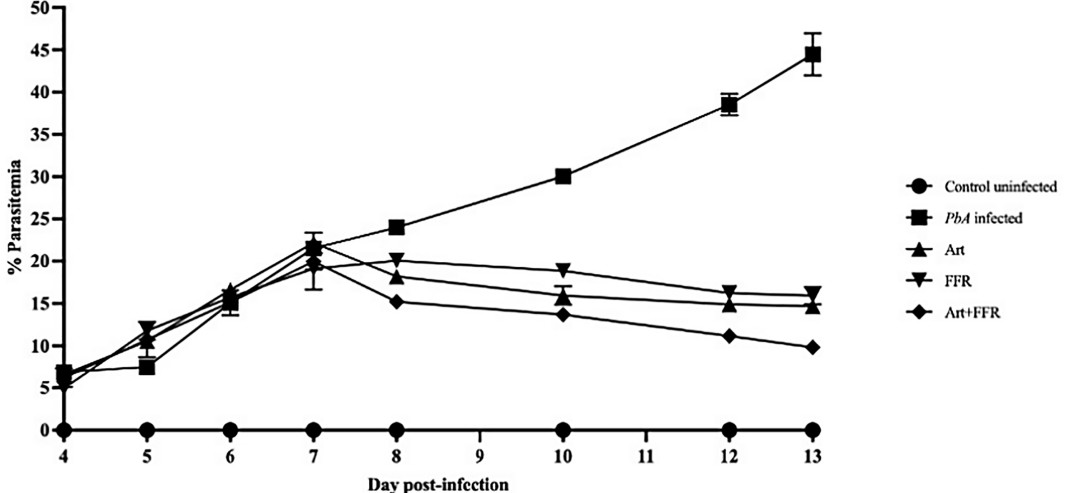

**Fig 2. Assessment of crude extracts from the five-flower remedy in an experimental cerebral malaria model.** C57BL/6 mice were inoculated with $1 \times 10^7$ *Plasmodium berghei* ANKA and administered artesunate, crude extract from the five-flower remedy, or 7% Tween 80 from days 6 to 12 post-infection. Parasitemia levels were assessed using thin blood smear films prepared from tail vein samples on days 4, 5, 6, 7, 8, 10, 12, and 13. The percentage of parasitemia was determined by counting infected red blood cells in at least 300 total red blood cells. Data are expressed as the mean±SEM (n=5/group), $p < 0.05$.



**Table 4. Effects of crude extract from the five-flower remedy on the percentage of parasite (day 6 before treatment and day 13 post-infection) and parasite suppression in experimental cerebral malaria at day 13 post-infection.**

| Groups | Dose (mg/kg) | % Parasitemia | | % Suppression (Day 13) |
|---|---|---|---|---|
| | | Day 6 | Day 13 | |
| *P. berghei* (*Pb*A) | – | 15.06 ± 1.45 | 44.47 ± 2.49[b, c, d] | – |
| Artesunate (Art) | 6 | 16.60 ± 0.37 | 14.68 ± 0.87[a, d] | 66.97 ± 1.95[d] |
| FFR alone | 600 | 15.72 ± 0.45 | 15.91 ± 1.02[a, d] | 64.22 ± 2.31[d] |
| Art + FFR | Art 6 + FFR 600 | 15.15 ± 0.57 | 9.81 ± 0.42[a,b, c] | 77.94 ± 1.86[b, c] |

Data are expressed as the mean ± SEM (n = 5/group), $p < 0.05$. [a]Significantly higher than that of the negative control. [b]Significantly higher than those of the groups treated with artesunate. [c]Significantly higher than those of the groups treated with FFR 600 mg/kg extract. [d]Significantly higher than those of the groups treated with Art + FFR 600 mg/kg extract.

During infection, *Pb*A-infected untreated mice exhibited a progressive decline in body weight, with an estimated reduction of −4.35% compared to their baseline weight, reflecting disease progression and the systemic effects of CM. By day 13 post-infection, mice treated with artesunate alone showed a moderate improvement in body weight, with an increase of 1.96% compared to their baseline weight. Similarly, mice receiving the ethanolic crude extract of FFR alone exhibited a weight increase of 1.07%, suggesting a partial mitigation of infection-induced weight loss. Notably, the combination therapy of artesunate and FFR resulted in the most significant improvement, with an average body weight increase of 2.80% compared to their baseline weight. Statistical analysis confirmed that the artesunate combined with the FFR group exhibited significantly higher body weight changes ($p < 0.05$), indicating a potential effect of the combined treatment in alleviating infection-related weight loss (Table 5).

### Quantitative evaluation of CM severity using the RMCBS scoring

To evaluate the severity of neurological impairments caused by CM, the RMCBS was used to assess mice at multiple time points: days 1, 4, 5, 6, 8, 10, and 12 post-infections (Fig 3). This scoring system allowed for a quantitative analysis of disease progression by measuring various behavioral and neurological parameters. By day 6 post-infection, *Pb*A-infected mice exhibited noticeable neurological deficits, reflected by a significant reduction in their RMCBS scores. These deficits included decreased exploratory behavior, signs of physical weakness, and impaired motor coordination, indicating the onset of severe CM symptoms. At day 12 post-infection, *Pb*A-infected untreated mice displayed the most severe neurological impairments, as evidenced by the lowest RMCBS scores. These mice exhibited inactive exploratory behavior, ruffled fur, abnormal reflex responses, muscle weakness, lack of escape responses, and convulsions. Such symptoms

**Table 5. Effect of crude extracts from five-flower remedy on the body weights of infected mice in an experimental cerebral malaria model.**

| Groups | Mean body weight (g) | | | % changes |
|---|---|---|---|---|
| | Day 0 | Day 6 | Day 13 | |
| Control uninfected | 20.36 ± 0.30 | 22.97 ± 0.46 | 23.69 ± 0.50 | 3.13 ± 0.52 [b] |
| *Pb*A-infected | 21.11 ± 0.30 | 22.22 ± 0.41 | 21.26 ± 0.53 | −4.35 ± 1.39 [a, c, d, e] |
| Art 6 mg/kg | 21.64 ± 0.26 | 22.48 ± 0.43 | 22.92 ± 0.47 | 1.96 ± 0.77 [b] |
| FFR 600 mg/kg | 20.70 ± 0.32 | 21.32 ± 0.32 | 21.61 ± 0.47 | 1.07 ± 0.91[b] |
| Art + FFR | 20.65 ± 0.40 | 22.04 ± 0.30 | 22.63 ± 0.23 | 2.80 ± 1.52 [b] |

Data are expressed as the mean ± SEM (n = 5/group), $p < 0.05$, with the following time points: day 0: day of infection; day 6: before treatment; and day 13: after completing treatment. [a]Significantly higher than that of the control uninfected group. [b]Significantly higher than that of the negative control group. [c]Significantly higher than those of the groups treated with artesunate. [d]Significantly higher than those of the groups treated with 600 mg/kg extract. [e]Significantly higher than those of the groups treated with Art + 600 mg/kg extract.

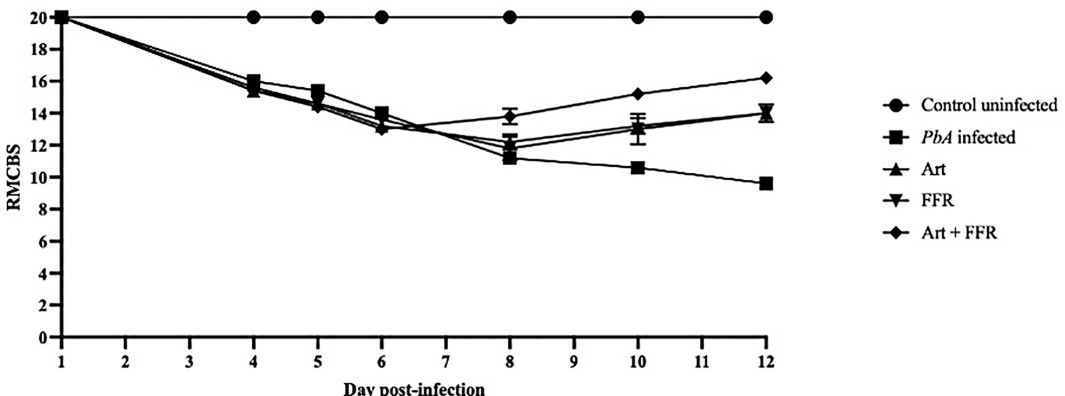

**Fig 3. Assessment of crude extracts from the five-flower remedy in an experimental cerebral malaria model.** C57BL/6 mice were inoculated with $1 \times 10^7$ *Plasmodium berghei* ANKA and administered artesunate, crude extract from the five-flower remedy, or 7% Tween 80 from days 6 to 12 post-infection. Clinical scores were recorded on days 1, 4, 5, 6, 8, 10, and 12 post-infections to evaluate disease progression and treatment efficacy. Data are expressed as the mean ± SEM (n = 5/group), $p < 0.05$.

are characteristics of advanced CM and indicate severe neurological dysfunction. Interestingly, RMCBS scores were significantly higher in the artesunate-treated group than in the untreated *Pb*A-infected group ($p < 0.05$). Furthermore, RMCBS scores were higher in *Pb*A-infected mice that received the ethanolic crude extract of FFR than in the untreated group ($p < 0.05$), indicating a potential neuroprotective role of FFR in CM. Notably, the combination therapy of artesunate and FFR yielded the most promising results, with treated mice displaying the highest RMCBS scores among all infected groups ($p < 0.05$).

### Cognitive dysfunction and long-term memory impairment in the ECM model

The NOR test was conducted on days 12–13 post-infection to evaluate memory retention and discrimination ability. This test measures the ability of an animal to recognize a previously encountered object, which is a well-established indicator of cognitive performance and long-term memory function. In the *Pb*A-infected-untreated group, mice exhibited significant impairments in NOR memory. These mice demonstrated a lower DI than that of the control group ($p < 0.05$), indicating a diminished ability to distinguish between familiar and novel objects. In contrast, *Pb*A-infected mice that received artesunate or the ethanolic crude extract of FFR alone showed improvements in recognition memory, as reflected by an increased DI compared to that of the *Pb*A-infected-untreated group (Fig 4). This suggests that both artesunate and FFR alone exert neuroprotective effects, potentially reducing the cognitive impairments associated with CM. Notably, *Pb*A-infected mice that received the combination therapy of artesunate and FFR exhibited the most significant improvement in cognitive performance. These mice demonstrated a markedly higher DI than that of the *Pb*A-infected-untreated group ($p < 0.05$) (Fig 4). These findings highlight the potential therapeutic benefits of adjunctive treatment in improving cognitive function and preventing long-term memory impairment in CM.

### Expression profiles of inflammatory cytokines and neurotrophic factors in brain tissue

On day 13 post-infection, qRT-PCR was performed to assess the gene expression of key inflammatory cytokines and neurotrophic factors in brain tissue samples. The analysis revealed a significant upregulation of tumor necrosis factor-alpha (TNF-α) (Fig 5A) and interleukin-1 beta (IL-1β) (Fig 5B) in the *Pb*A-infected untreated group, compared to the uninfected control group ($p < 0.05$), indicating a heightened inflammatory response associated with infection. Treatment with artesunate combined with FFR significantly downregulated TNF-α expression compared to the *Pb*A-infected untreated group

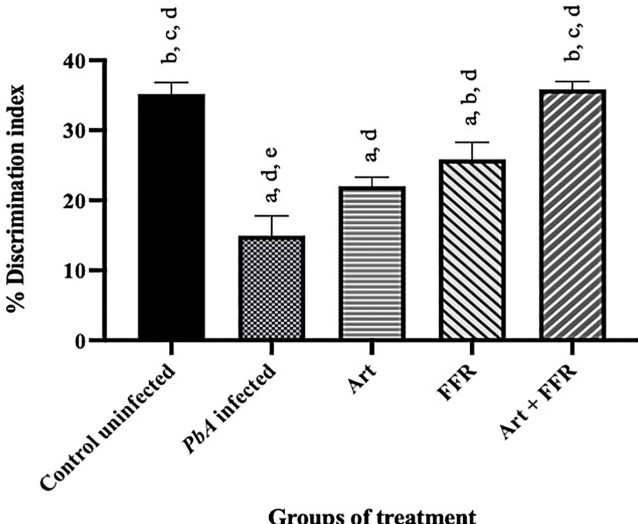

**Fig 4. Effect of crude extracts from the five-flower remedy on cognitive dysfunction and long-term memory impairment in experimental cerebral malaria model using novel object recognition test.** Mice were subjected to a novel object recognition test on day 12 post-infection for the habituation and training sessions and on day 13 post-infection for the testing session. The percentage of discrimination index between the groups was recorded. *PbA, Plasmodium berghei* ANKA; Art, artesunate; FFR, five Fflower remedy: NOR: novel object recognition. Results are shown as the mean ± SEM (n = 5/group), $p < 0.05$. [a]Significantly higher than that of the control uninfected group. [b]Significantly higher than that of the negative control group. [c]Significantly higher than those of the groups treated with artesunate. [d]Significantly higher than those of the groups treated with 600 mg/kg extract. [e]Significantly higher than those of the groups treated with Art + 600 mg/kg extract.

($p < 0.05$). Furthermore, IL-1β expression was significantly reduced in all treatment groups—including artesunate alone, FFR alone, and the combined treatment group—when compared to the *Pb*A-infected untreated group ($p < 0.05$), suggesting that each intervention contributed to attenuating the neuroinflammatory response.

In contrast to the cytokine expression profile, the expression of neurotrophic markers showed a reverse pattern. The gene expression of brain-derived neurotrophic factor (BDNF) (Fig 5C) and its receptor, tropomyosin receptor kinase B (TrkB) (Fig 5D), was markedly increased in all treated groups (artesunate alone, FFR alone, and combination therapy) compared to the *Pb*A-infected untreated group ($p < 0.05$). Conversely, a significant downregulation of both BDNF and TrkB was observed in the *Pb*A-infected untreated group relative to all treatment groups ($p < 0.05$). These findings indicate that the therapeutic interventions not only mitigated inflammation but also promoted neurotrophic signaling pathways that may contribute to neuroprotection and functional recovery following infection.

## Morphological changes in the brain tissues

At 13 days post-infection, brain tissue analysis of the control group revealed no significant alterations in the cellular parenchyma. No evidence of vascular dilation or capillary obstruction was observed (Fig 6A), and no signs of brain hemorrhage were detected (Fig 6B). In addition, the neurons in the hippocampus region remained well-preserved (Fig 6C), indicating normal brain structure and function in this group. In contrast, brain sections from the *Pb*A-infected-untreated group exhibited marked pathological changes. A significant accumulation of parasitized red blood cells and leukocytes was observed within the brain vessels (Fig 6D), suggesting severe vascular impairment. Moreover, multiple hemorrhagic areas were detected in the cerebral parenchyma (Fig 6E), indicating widespread damage caused by the infection. However, these histopathological alterations were notably reduced in *Pb*A-infected mice treated with either artesunate or FFR alone. In cortical brain sections that initially displayed severe inflammation, mice treated with artesunate (Fig 6G) or FFR alone (Fig



**Fig 5. Comparative analysis of the effects of five-flower remedy crude extract on the expression of inflammatory cytokines and neurotrophic factors in the brain of *Pb*A-infected mice.** *Plasmodium berghei* ANKA (PbA)-infected mice were administered five-flower remedy (FFR), and quantitative real-time PCR (qRT-PCR) was performed on brain tissues on day 13 post-infection. mRNA levels of inflammatory cytokines (A: TNF-α; B: IL-1β) and neurotrophic factors (C: BDNF; D: TrkB) were quantified. Gene expression in the *Pb*A-infected untreated, artesunate-treated (Art), FFR-treated, and combination treatment (Art+FFR) groups are presented relative to the uninfected control group. Expression data were normalized to GAPDH as the internal reference gene, and relative quantification was calculated using the comparative threshold cycle (ΔΔCt) method. Data are presented as mean±standard error of the mean (SEM) (n=5 per group), $p < 0.05$ and representative of at least two independent experiments. *Pb*A, *Plasmodium berghei* ANKA; Art, artesunate; FFR, five-flower remedy. [a]Significantly higher than that of the control uninfected group. [b]Significantly higher than that of the negative control group. [c]Significantly higher than those of the groups treated with artesunate. [d]Significantly higher than those of the groups treated with 600 mg/kg extract. [e]Significantly higher than those of the groups treated with Art+600 mg/kg extract.

6J) exhibited reduced cellular infiltration, smaller hemorrhagic areas (Fig 6H and 6K), and fewer occluded capillaries compared to the untreated *Pb*A-infected group. Furthermore, brain sections from *Pb*A-infected mice that received the combined artesunate and FFR treatment (Fig 6M) showed only mild lesions. These sections also contained fewer parasitized



**Fig 6. Protective effects of crude extracts from the five-flower remedy on brain histopathology in an experimental cerebral malaria model.**
C57BL/6 mice were inoculated with *Plasmodium berghei* ANKA (PbA)-infected erythrocytes and treated with crude extracts from the five-flower remedy for 7 consecutive days. Brain histological sections were analyzed across different groups: uninfected control (**A**, **B**, **C**), *Pb*A-infected untreated (**D**, **E**, **F**), artesunate-treated (**G**, **H**, I), five-flower remedy alone treated (**J**, **K**, **L**), and a combination of artesunate with the five-flower remedy (**M**, **N**, **O**). The uninfected control group exhibited normal cerebral cortex histology with healthy neurons and intact blood vessels **(A)**, no brain hemorrhage **(B)**, and well-preserved neurons in the hippocampal region **(C)**. In contrast, the infected untreated group showed a high accumulation of parasitized red blood cells (**D**, arrowhead) and extensive hemorrhagic areas (**E**, yellow arrow). Treatment with artesunate **(H)**, the five-flower remedy **(K)**, and their combination (N) resulted in reduced hemorrhagic regions. The hippocampal coronal sections (CA1, CA3, and dentate gyrus) revealed neuronal loss in the infected untreated group (**F**, black box), whereas progressive neuronal recovery was observed in the artesunate **(I)**, five-flower remedy alone **(L)**, and combination treatment groups **(O)**. *PbA, Plasmodium berghei* ANKA; Art, artesunate; FFR, five flower remedy. The magnifications are 10X with a scale bar = 200 µm for panels **C**, **F**, **I**, **L**, and **O**, 20X with a scale bar = 200 µm for panels **B**, **E**, **H**, **K**, and **N**, and 40X with a scale bar = 100 µm for panels **A**, **D**, **G**, **J**, and **M.**

red blood cells and leukocytes obstructing the brain vessels and showed a small focus on brain hemorrhage (Fig 6N), indicating the protective effect of the combined treatment in mitigating infection-related damage.

The hippocampal structure was analyzed to identify pathological changes. In the control (uninfected) group, pyramidal and granular cells were arranged in an orderly, densely packed manner, maintaining their normal structure (Fig 6C). Hematoxylin and eosin staining (Fig 6F) showed a marked increase in neurodegeneration in the hippocampus of *Pb*A-infected mice compared to the control group. The hippocampal tissue of *Pb*A-infected mice exhibited neuronal loss, neuronal shrinkage, and cytoplasmic vacuolar degeneration. Conversely, mice treated with artesunate (Fig 6I), FFR alone (Fig 6L), or a combination of artesunate and FFR (Fig 6O) showed better-preserved hippocampal morphology. Their pyramidal cell layers remained well-organized, with fewer neurons displaying abnormal morphology compared to the control uninfected group.

## Discussion

CM is a life-threatening neurological syndrome characterized by unregulated immune responses and substantial disruptions to the central nervous system. Numerous studies have documented its deleterious effects on the brain, with cognitive dysfunction emerging as a common long-term neurobehavioral consequence among survivors [41]. This process is aggravated by excessive inflammation, leading to neuronal damage and brain swelling (cerebral edema) [42]. Moreover, neuroinflammatory processes, including the overproduction of pro-inflammatory cytokines, play a pivotal role in mediating neuronal injury and cognitive impairment in CM. Although animal models of CM do not fully replicate the complexity of human CM, they share key neuropathological features, such as hyperinflammation, intravascular accumulation of immune T cells, BBB breakdown, brain swelling, hemorrhages, and neuronal damage caused by infected RBC sequestration [41,43]. Despite advancements in anti-malarial treatment, including the widespread use of artesunate, CM remains associated with high mortality rates and long-term neurological sequelae. Artesunate, a potent and fast-acting artemisinin derivative, has significantly improved survival outcomes; however, it does not fully prevent neurological impairments [18]. Previous studies have reported that high doses of artemisinin derivatives (artemether or artesunate), ranging from 25 to 50 mg/kg, resulted in an unexpected pattern of neuronal damage [44]. Moreover, recent reports indicate a decline in the effectiveness of artemisinin-based combination therapy partner drugs and a growing tolerance to artemisinin derivatives, highlighting the urgent need for new antimalarial compounds with different mechanisms of action. Therefore, developing novel drug molecules or combinations with an extended half-life, an optimal effective dose (ED50) for effectively reducing parasite load, and anti-inflammatory properties is crucial. In addition, these drugs should be minimally toxic and safe for clinical use [45,46].

Natural products have historically served as a rich source of bioactive compounds with considerable pharmacotherapeutic potential. A diet abundant in natural products is widely recognized for its protective effects against various diseases, including cardiovascular disorders, metabolic syndromes, and neurodegenerative conditions [47]. Notably, numerous phytochemicals exhibit neuroprotective properties by targeting oxidative stress, neuroinflammation, and mitochondrial dysfunction—key contributors to neuronal damage in CM. Studies have shown that natural compounds such as flavonoids, alkaloids, and terpenoids can modulate critical signaling pathways involved in neurodegeneration, offering therapeutic promise for conditions such as Parkinson's disease, Alzheimer's disease, and ischemic stroke [48–50]. In addition, bioactive molecules derived from medicinal plants have demonstrated efficacy against a range of infectious diseases, including leishmaniasis and malaria [51,52]. Given their diverse pharmacological properties, natural products represent a promising avenue for developing novel anti-malarial agents with neuroprotective capabilities [53–55]. Integrating bioactive phytochemicals into existing malaria treatment regimens may enhance therapeutic outcomes, mitigate neuronal damage, and improve long-term cognitive recovery in CM survivors.

Our previous study demonstrated that the ethanolic extracts of FFR exhibited significant *in vitro* anti-plasmodial activity against the *P. falciparum* K1 strain, with an $IC_{50}$ of $2.8 \pm 0.3$ μg/mL. In addition, the extracts showed minimal

cytotoxicity towards the Vero cell line, as indicated by a selectivity index > 2, suggesting a favorable safety profile [24]. In *in vivo* experiments, treatment with FFR at a dose of 600 mg/kg resulted in a notable parasitemia suppression rate of 61.49 ± 1.57. Furthermore, the $LD_{50}$ values of these extracts exceeded 2,000 mg/kg, confirming their safety in an acute oral toxicity assessment conducted in a murine model (unpublished data). Based on its anti-malarial potential, the FFR has been associated with a broad spectrum of pharmacological properties, including antioxidant and anti-inflammatory effects [25,26]. Given its promising anti-malarial efficacy observed in both *in vitro* and *in vivo* studies, along with a low toxicity profile, the FFR was selected for further investigation in this study as a potential therapeutic agent for ECM. In this study, the crude extracts from the FFR significantly reduced parasitemia, with the highest suppression rate observed in infected mice receiving the combination treatment of artesunate and FFR (Art-FFR), achieving approximately an 80% reduction. This outcome highlights the potential combination effect between artesunate, a well-established anti-malarial agent, and the bioactive compounds present in the FFR. Artesunate is known for its potent and rapid schizonticidal activity, directly targeting *Plasmodium* parasites and effectively reducing parasitemia levels [56]. Furthermore, our findings align with those from previous studies demonstrating the efficacy of natural plant extracts in suppressing parasitemia in murine models of CM. For instance, extracts from *Zizyphus spina-christi*, *Terminalia albida*, *Azadirachta indica*, and Cannabidiol can effectively reduce parasitemia in mouse models of CM [22,53,55,57]. These crude extracts exhibited schizonticidal activity in *Pb*A-infected mice. Their antimalarial effects are likely attributed to the presence of bioactive secondary metabolites, such as flavonoids, polyphenols, alkaloids, terpenoids, and saponins [58]. The observed anti-malarial effects of the FFR in *Pb*A-infected mice suggest that these extracts may exert schizonticidal activity, thereby interfering with the life cycle of the parasite and limiting its replication within the host. The pharmacological activity of these crude extracts is likely attributed to the presence of diverse secondary metabolites with well-documented anti-malarial properties [58]. These compounds may contribute to the observed suppression of parasitemia by exerting direct parasiticidal effects, modulating host immune responses, or enhancing the efficacy of artesunate when used in combination therapy.

Overall, the results of this study provide further evidence supporting the potential application of the FFR as a complementary therapeutic strategy for malaria. The main constituents of FFR and it components include alkaloids, triterpene, sesquiterpene, tannin, flavonoid, and coumarin [26]. These findings align with those from previous studies indicating that various secondary metabolites—including alkaloids, terpenes, flavonoids, xanthones, anthraquinones, phenols, and sesquiterpenes—exhibit antimalarial properties [59]. Terpenes and sesquiterpenes, naturally occurring compounds found in various plants, have demonstrated significant antimalarial properties through multiple mechanisms. In a mouse model infected with *P. berghei* ANKA, oral administration of nerolidol resulted in over 99% inhibition of parasitemia until 14 days post-infection, with a 90% survival rate observed on day 30, compared to 16% observed in controls [60]. Compounds such as limonene and linalool interfere with the biosynthesis of essential isoprenoids in *P. falciparum*, thereby inhibiting parasite development. Terpenes disrupt the elongation of isoprenic chains, leading to reduced synthesis of dolichol and ubiquinones, which are vital for parasite survival [61]. Furthermore, terpene forms coordination complexes with heme iron, preventing the sequestration of free heme into non-toxic forms such as beta-hematin. This disruption of heme detoxification is detrimental to the survival of the malaria parasite [62]. Alkaloids like quinine have antimalarial effects via preventing protein synthesis and interfering with the parasite's heme detoxification processes, among other ways. Quinine specifically prevents harmful free heme, which is produced during hemoglobin digestion, from polymerizing into hemozoin, an inactive crystalline form found in the parasite's feeding vacuole. In the end, this buildup of cytotoxic heme affects the life of the parasite and aids in its removal from the host [63,64]. Tannins contribute to antimalarial activity primarily through their potent antioxidant properties, which enable them to scavenge and neutralize reactive oxygen species (ROS), thereby reducing oxidative stress within infected erythrocytes. This redox-modulating effect may impair key physiological processes of the parasite, including heme polymerization [65]. In parallel, coumarin compounds have been shown to exert their antiparasitic effects by inhibiting the activity of DNA gyrase, a type II topoisomerase essential for maintaining DNA

topology. Inhibition of this enzyme results in the accumulation of single-stranded DNA breaks, thereby disrupting critical processes such as replication and transcription. In the context of *Plasmodium* species, DNA gyrase is localized within the apicoplast. Disruption of DNA gyrase function within the apicoplast impairs its self-replication and overall integrity. As a consequence, the parasite is unable to successfully complete its subsequent replication cycle, ultimately leading to cell death [66]. From previous study, Ostruthin, a naturally occurring coumarin derivative, was isolated from the root and stem extracts of *Luvunga sarmentosa*. This compound demonstrated antimalarial activity, exhibiting an $IC_{50}$ value of $2.65\pm0.07$ µg/mL against *Plasmodium falciparum*, the parasite responsible for the most severe form of human malaria. To further elucidate its potential mechanism of action, molecular docking analyses were performed. The results revealed that ostruthin displayed high binding affinities toward two critical parasitic enzymes: *Plasmodium falciparum* dihydroorotate dehydrogenase (PfDHODH) and lactate dehydrogenase (PfLDH), with calculated binding energies of $-9.94\pm0.11$ and $-8.84\pm0.11$ kcal/mol, respectively [67]. A flavonoid compound belonging to the dihydrochalcone subclass was successfully isolated from *Artocarpus altilis* and structurally characterized as 1-(2,4-dihydroxyphenyl)-3-[8-hydroxy-2-methyl-2-(4-methyl-3-pentenyl)-2H-1-benzopyran-5-yl]-1-propanone (hereafter referred to as Compound-1). Evaluation of its antimalarial activity demonstrated potent inhibition against *Plasmodium falciparum*, with an $IC_{50}$ value of 1.05 µM, indicating strong efficacy at low micromolar concentrations. Furthermore, an in silico molecular docking analysis was performed to explore its potential mechanism of action. The results suggested that Compound-1 exhibits high binding affinity toward the 3BPF receptor, a cysteine protease associated with falcipain-2, a key enzyme involved in hemoglobin degradation within the parasite. In conclusion, ethanolic extracts of the FFR as monotherapy inhibited parasitemia progression, whereas a combination of artesunate and crude extracts from the FFR significantly exerts a potent synergistic antimalarial effect. This observation demonstrated the possible combined effect of artesunate and crude extracts from FFR, as artesunate is rapidly absorbed in the tissues and quickly excreted after administration. Furthermore, a previous study suggested that the crude extracts may enhance the bioactivity of artesunate and improve its permeability in infected red blood cells. Combination therapy is highly effective, rapidly eliminating parasites and reducing fever within 48 h in 99% of patients [68]. This rapid parasite clearance helps prevent early-phase recrudescence. In contrast, monotherapy may result in slower parasite elimination and an increased risk of recrudescence [69]. The fast and nearly complete parasite clearance achieved with artemisinin-based combination therapies may also prevent parasite sequestration in the brain, a key factor in reducing the severity of CM. Therefore, the rapid and almost complete parasite clearance may prevent parasite sequestration in the brain. The findings of this study demonstrate that the ethanolic extracts of the FFR, when administered as monotherapy, effectively inhibit the progression of parasitemia. However, a more pronounced and synergistic anti-malarial effect was observed when these extracts were combined with artesunate. This significant enhancement in therapeutic efficacy suggests a potential interaction between artesunate and the bioactive compounds present in the FFR. Studies have proposed that crude plant extracts, including those from the FFR, may enhance the bioactivity of artesunate by increasing its stability, prolonging its bioavailability, and improving its permeability into infected red blood cells [46]. This improved intracellular uptake likely facilitates more effective parasite clearance and contributes to the observed synergistic anti-malarial activity. These findings underscore the therapeutic potential of integrating plant-based bioactive compounds into conventional anti-malarial regimens. Moreover, artesunate is rapidly converted into its active metabolite, dihydroartemisinin (DHA), predominantly through plasma esterases and hepatic metabolic enzymes, including cytochrome P450 isoforms (notably CYP2A6) and UDP-glucuronosyltransferases (UGTs) [70]. Phytoconstituents in FFR—particularly flavonoids, alkaloids, and phenolic acids—are known to influence the activity of these metabolic pathways as well as drug transport proteins such as P-glycoprotein. The observed between artesunate and the FFR suggests that natural extracts may serve as effective adjuncts to existing anti-malarial therapies, enhancing drug efficacy while potentially mitigating adverse effects [71]. Future studies should focus on elucidating the precise mechanisms underlying the anti-malarial activity of its bioactive constituents, optimizing its formulation, and conducting comprehensive pharmacokinetic and toxicity assessments to facilitate its potential clinical translation.

CM is characterized by a range of cellular and histopathological changes, including the sequestration of parasitized erythrocytes in cerebral microvasculature, widespread neuroinflammation, and the disruption of BBB. These processes likely trigger cell-mediated apoptosis, leading to the death of vulnerable neuronal populations [72]. Such pathological mechanisms contribute to severe neurological dysfunction, often leading to significant behavioral impairments in infected animals [73]. ECM serves as a valuable tool for investigating potential therapeutic interventions, as it replicates many of the clinical and pathological features observed in human CM [74]. In this study, *Pb*A-infected mice exhibited hallmark signs of CM, including severe muscle weakness, abnormal gait patterns, impaired reflexes, rolling behavior, and convulsions, particularly around day 6 post-infection. These neurological symptoms are consistent with previous observations in ECM models. However, mice treated with FFR monotherapy demonstrated a significant improvement in neurological outcomes, as evidenced by an increase in RMCBS score and only mild signs of neurological impairment. Notably, the combination therapy of artesunate and FFR led to a remarkable enhancement in survival rates, with treated mice surviving up to 13 days post-infection without exhibiting any observable neurological deficits. These findings align with those from previous studies, indicating that an improvement in survival rate is closely associated with better neurological parameters [75,76]. The observed neuroprotective effects may be attributed to the various bioactive phytochemicals present in the crude extracts of the FFR. Notably, these include fatty acid (linolenic acid, linoleic acid, ethyl hexadecanoate, and oleamide), triterpene (squalene and alpha-kaurene), sesquiterpene (alloaromadendrene, delta-cadinene, beta-caryophyllene, gamma-cadinene, aromandendrene, gamma-muurolene, and humulene), and the heterocyclic compound 2-Hydroxy-1,8-naphthyridine. Linoleic acid helps lower inflammation by decreasing the secretion and expression of key cytokines (IL-6, TNF-α, and IL-1β) and chemokines (IL-8, CCL2, and RANTES), which are central to the "cytokine storm." In addition, it helps regulate the immune system by reducing the release of inflammatory cytokines from macrophages and encourages macrophages to shift toward an M2 anti-inflammatory type rather than the pro-inflammatory M1 type. Furthermore, linoleic acid decreases neutrophil infiltration into tissues, overall inflammation, and circulating monocyte levels. In oxidative stress protection, it reduces oxidative stress by lowering reactive oxygen species (ROS) levels in situations where excessive oxidation occurs [77]. Studies have demonstrated that oleamide suppresses inflammatory responses in lipopolysaccharide-induced RAW264.7 murine macrophages and alleviates paw edema in a carrageenan-induced inflammatory rat model. These results indicate that oleamide exhibits anti-inflammatory effects, potentially through the inhibition of NF-κB activation [78]. Another study showed that oleamide suppresses microglial TNF-α production in a concentration-dependent manner. Oleamide may enhance microglial anti-inflammatory activity, which could be beneficial in neuroinflammatory conditions by reducing the accumulation of amyloid β and hippocampal inflammation (TNF-α and MIP-1α production) while enhancing hippocampal neurotrophic factors (BDNF and GDNF) [79]. Triterpenes from *Ganoderma lucidum* have demonstrated potent antioxidant properties by scavenging free radicals, enhancing antioxidant enzyme activity, and protecting against oxidative-stress-induced protein and lipid peroxidation in liver and brain tissues of aged mice [80]. Sesquiterpenes have shown promise in protecting against neurodegenerative diseases. Nerolidol, a sesquiterpene alcohol found in essential oils from plants, has exhibited neuroprotective effects by mitigating neuroinflammation and oxidative stress —two key contributors to neurodegeneration. This suggests that sesquiterpenes may play a role in preserving neuronal health and function [81]. A novel 1,8-naphthyridine-2-carboxamide derivative has exhibited significant anti-inflammatory activity. In lipopolysaccharide-treated BV2 microglial cells, this compound reduced the production of pro-inflammatory cytokines and suppressed the Toll-like receptor 4/myeloid differentiation factor 88/NF-κB signaling pathway, suggesting its potential to attenuate inflammatory responses by modulating key inflammatory pathways. Furthermore, 1,8-Naphthyridine derivatives have shown potential applications in neurological disorders, including Alzheimer's disease, multiple sclerosis, and depression. Their ability to interact with various biological targets makes them promising candidates for the development of therapeutic agents aimed at treating neurodegenerative diseases [82]. These compounds modulate key signaling pathways involved in oxidative stress, inflammation, and immune response regulation, which are crucial factors in CM pathogenesis. The ability of these phytochemicals to mitigate neuroinflammation and

oxidative damage suggests that their presence in the FFR may contribute to reducing disease severity, preserving neurological function, and improving overall survival in ECM. By alleviating inflammatory damage, these bioactive compounds could play a crucial role in counteracting the devastating neurological consequences of CM. In general, the results of this study provide compelling evidence that integrating natural bioactive compounds into anti-malarial treatment regimens holds promise for enhancing both survival and neurological outcomes in CM.

The elevated expression of TNF-α and IL-1β observed in this study aligns with previous reports highlighting the pivotal role of pro-inflammatory cytokines in CM pathogenesis [83,84]. TNF-α exacerbates BBB permeability by altering tight junction integrity and promoting the expression of adhesion molecules such as ICAM-1 and VCAM-1, leading to the sequestration of parasitized erythrocytes and leukocytes in cerebral microvessels—a hallmark of CM pathology [85]. In addition, IL-1β, through its role in glial activation and induction of NO and ROS, contributes to oxidative stress and neuronal apoptosis, further amplifying the inflammation within the brain [86]. The therapeutic reduction in pro-inflammatory gene expression following treatment with artesunate and/or FFR highlights their potential role in modulating the host immune response and attenuating the neuroinflammatory burden. Although artesunate remains the first-line treatment for severe malaria owing to its rapid parasiticidal action, recent studies have shown that it also exerts immunomodulatory effects, likely through NF-κB pathway suppression, reduction of pro-inflammatory cytokine production, and attenuation of leukocyte-endothelial interactions [87]. Similarly, the FFR, a traditional polyherbal formulation, may offer adjunctive neuroprotective benefits, possibly through its antioxidant, anti-inflammatory, and anti-apoptotic constituents, which have demonstrated efficacy in models of neurodegenerative and neuroinflammatory disorders [88,89]. The neuroprotective effects observed in the FFR-treated group may be attributed to multiple overlapping mechanistic pathways. Firstly, the antioxidant activity of FFR is likely to play a crucial role in mitigating oxidative stress-induced neuronal injury. Several constituent herbs in FFR including *Syzygium malaccense*, and *Tulipa edulis* have been reported to contain high levels of flavonoids with strong free radical scavenging capacity [90,91]. Flavonoids can donate a hydrogen atom to neutralize free radicals. Also, flavonoids may act by single-electron transfer [92]. By reducing the generation of ROS, FFR may protect neuronal structures from oxidative damage commonly associated with cerebral malaria. Secondly, FFR may exert anti-inflammatory effects that contribute to neuroprotection. Previous studies have shown that certain components of FFR can downregulate pro-inflammatory cytokines such as TNF-α and IL-1β, which are involved in the pathogenesis of neuroinflammation during malaria [83]. Suppression of neuroinflammatory signaling pathways may reduce neuronal apoptosis and improve neurological outcomes. Thirdly, stabilization of the BBB may represent another neuroprotective mechanism of FFR. The integrity of the BBB is often compromised during cerebral malaria, facilitating infiltration of immune cells and neurotoxic substances into the brain parenchyma. Certain phenolic plant extracts have been shown to enhance tight junction protein expression including ZO-1, claudin-5, and occludin, thereby preserving BBB integrity under pathological conditions [93].

Moreover, the restoration of neurotrophic signaling, as evidenced by significant upregulation of BDNF and its receptor TrkB, offers further insight into the neuroregenerative potential of these treatments. BDNF, a critical neurotrophin involved in neuronal survival, synaptic plasticity, axonal growth, and memory formation, is frequently suppressed under conditions of chronic inflammation and oxidative stress [94]. The increase in BDNF and TrkB expression following treatment suggests a favorable shift toward neuroprotective and repair-promoting pathways, which may counteract the neurocognitive impairments and behavioral deficits commonly observed in CM survivors. Notably, CM is associated with long-term neurological sequelae, including cognitive dysfunction, attention deficits, motor impairments, and behavioral abnormalities, particularly in pediatric populations [95]. Therefore, interventions that not only eliminate the parasite but also mitigate neuroinflammation and promote neuronal recovery are of immense clinical relevance. The ability of the artesunate and FFR combination therapy to simultaneously suppress inflammatory mediators and restore neurotrophic signaling pathways suggests a holistic therapeutic approach capable of improving both acute survival and long-term neurological outcomes in CM. Furthermore, these findings support the emerging paradigm that targeting host-pathogen interactions, and the inflammatory environment is as critical as antiparasitic efficacy in managing severe malaria. Adjunctive therapies such as the

FFR may represent a promising avenue for integrated neuroprotective strategies, particularly in resource-limited settings where post-malaria care and neurological rehabilitation are often inaccessible [8].

Behavioral and cognitive alterations associated with CM are closely linked to BBB dysfunction. Multiple studies have established that *P. falciparum* infection can trigger long-term neurological impairments, including deficits in learning and memory. The underlying mechanisms contributing to these impairments are largely attributed to neuroinflammation and oxidative stress [36]. During disease progression, excessive activation of the immune response leads to the overproduction of pro-inflammatory cytokines and ROS, which not only exacerbate neuronal damage but also interfere with neurotrophic factors. These neurotrophic factors play a crucial role in neuronal survival, synaptic plasticity, and adult neurogenesis—processes vital for cognitive function and memory formation [96]. Neurotrophic factors, such as BDNF, are crucial for neuronal survival, differentiation, and synaptic plasticity. Chronic inflammation and oxidative stress can downregulate the expression of these factors, hindering neurogenesis and synaptic remodeling. Reduced levels of BDNF have been observed in patients with neurodegenerative disorders, correlating with cognitive deficits and disease progression [97]. In addition to human studies, several experimental models of malaria have demonstrated significant cognitive and behavioral alterations in infected animals [98]. In this study, *Pb*A-infected mice exhibited progressive long-term memory deficits, reinforcing previous findings that ECM can lead to substantial neurocognitive impairments. However, the administration of ethanolic extracts from the FFR effectively mitigated these behavioral deficits. Notably, the combination therapy of artesunate and FFR resulted in a progressive and statistically significant improvement in cognitive function when compared to the *Pb*A-infected untreated group. These findings are consistent with those of previous studies demonstrating that co-administration of standard anti-malarial drugs with adjunctive neuroprotective therapies can improve cognitive outcomes [98]. One of the key factors contributing to the observed neuroprotection is the impact of the FFR on BBB integrity and inflammatory response. BBB breakdown is a major driver of neuroinflammation and subsequent behavioral alterations in CM. The inflammation created during infection facilitates the influx of activated immune cells, including monocytes and T cells, into brain tissue, thereby exacerbating neuronal damage [69]. The crude extracts from the FFR exhibit significant anti-inflammatory effects, which may help restore BBB integrity and prevent excessive neuroinflammatory responses. Specifically, these bioactive compounds downregulate pro-inflammatory cytokines and suppress the activation of microglia—key mediators of neuroinflammation. By modulating these pathways, the FFR can protect against CM-associated neurodegeneration and improve cognitive function.

Beyond its neuroprotective effects, this study provides compelling evidence that the combination of artesunate and the FFR significantly enhances anti-malarial efficacy. This is the first comprehensive set of studies demonstrating the synergistic anti-*Plasmodium* effects of crude extracts when used with artesunate against ECM induced by *P. berghei*. The combined treatment regimen resulted in a rapid clearance of parasites without recrudescence during the early phase of infection. Specifically, the combination therapy achieved an 80% chemosuppression rate, which was significantly higher than the individual suppression rates observed with either treatment alone (approximately 60%). The enhanced parasite clearance observed in the combination therapy group is particularly relevant for preventing severe CM pathology. One of the most devastating consequences of malaria infection is the sequestration of *Plasmodium*-infected erythrocytes within the cerebral microvasculature, leading to vascular occlusion, ischemia, and inflammation-driven neuronal damage. The rapid elimination of parasitemia in the artesunate combined with the FFR-treated group suggests that this combination therapy may effectively prevent parasite sequestration, thereby reducing CM-associated neurovascular damage. CM survivors often experience long-term cognitive deficits, motor dysfunction, and psychiatric disorders due to prolonged exposure to parasite-derived neurotoxins and inflammatory mediators. Furthermore, our results indicate that the combination treatment is associated with a well-regulated interplay of inflammatory cytokines and neurotrophic markers. The rapid parasite elimination observed in the artesunate combined with the FFR-treated group suggests that this combination therapy effectively prevents parasite sequestration, thereby reducing the likelihood of CM-associated neurovascular damage. Beyond its remarkable anti-malarial efficacy, the artesunate and FFR combination exhibited profound immune-modulating

properties. The study uncovered a well-regulated interplay between inflammatory cytokines, which are crucial in controlling the immune response of the host to malaria infection. By balancing immune activity, this combination treatment not only enhanced survival rates but also significantly reduced the long-term neurocognitive impairments that plague CM survivors. These immune-modulating effects likely contribute to improved clinical outcomes, reduced mortality rates, and a significant reduction in long-term neurocognitive impairments following infection. The ability of the FFR to modulate inflammatory responses while concurrently enhancing artesunate efficacy highlights its potential as a promising adjunctive therapy for CM. Despite the promising findings, the extract used in this study was a crude preparation, and the specific bioactive compounds responsible for the observed effects were neither isolated nor quantified. Furthermore, long-term safety, and pharmacokinetic profiles were not assessed, limiting the ability to conclusions regarding the extract's suitability for chronic administration or adjunctive use in clinical settings. These limitations highlight the need for further comprehensive investigations, including studies on potential drug interactions, isolation and characterization of active constituents, elucidation of underlying mechanisms of action, and clinical trials to assess the safety and efficacy of this combination therapy in human populations at risk of CM.

## Conclusions

Our findings indicate that the FFR holds promise as a potential adjuvant therapy (in combination with antimalarial drugs) to prevent brain damage and neurological complications associated with CM in mice. The oral co-administration of artesunate and the FFR demonstrated antimalarial activity, anti-inflammatory effects, and significant improvements in histopathological changes induced by *Pb*A infection. In addition, this combined treatment preserved cognitive function and inhibited the progression of CM. In this context, co-administration effectively protected against the development of ECM and could serve as a potential alternative therapeutic strategy for treating CM. However, further studies are required to elucidate the drug interaction and precise mechanisms underlying the effects of the FFR.

## Supporting information

**S1 File. Data tables for graph generation.**
(PDF)

**S2 File. Brain histopathology in an experimental cerebral malaria (ECM) Model.**
(PDF)

**S3 File. Melt curve data for qPCR.**
(PDF)

## Acknowledgments

The authors express gratitude to the Center for Scientific and Technological Equipment of Walailak University for their valuable support and assistance in providing laboratory equipment and facilities for this study. In addition, we appreciate the staff at the Department of Tropical Pathology, Faculty of Tropical Medicine, Mahidol University, Thailand, for their assistance with histological preparations and staining.

## Author contributions

**Conceptualization:** Walaiporn Plirat, Prapaporn Chaniad, Arisara Phuwajaroanpong, Atthaphon Konyanee, Laddawan Lalert, Chuchard Punsawad.

**Data curation:** Walaiporn Plirat, Prapaporn Chaniad, Arisara Phuwajaroanpong, Atthaphon Konyanee, Laddawan Lalert, Abdi Wira Septama, Chuchard Punsawad.



**Formal analysis:** Walaiporn Plirat, Prapaporn Chaniad, Arisara Phuwajaroanpong, Atthaphon Konyanee, Laddawan Lalert, Chuchard Punsawad.

**Funding acquisition:** Walaiporn Plirat, Prapaporn Chaniad, Chuchard Punsawad.

**Investigation:** Walaiporn Plirat, Prapaporn Chaniad, Arisara Phuwajaroanpong, Atthaphon Konyanee, Laddawan Lalert, Chuchard Punsawad.

**Methodology:** Walaiporn Plirat, Prapaporn Chaniad, Arisara Phuwajaroanpong, Atthaphon Konyanee, Laddawan Lalert, Chuchard Punsawad.

**Project administration:** Prapaporn Chaniad, Chuchard Punsawad.

**Writing – original draft:** Walaiporn Plirat, Prapaporn Chaniad, Arisara Phuwajaroanpong, Atthaphon Konyanee, Chuchard Punsawad.

**Writing – review & editing:** Walaiporn Plirat, Prapaporn Chaniad, Arisara Phuwajaroanpong, Atthaphon Konyanee, Laddawan Lalert, Abdi Wira Septama, Chuchard Punsawad.

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
