## [Decision Letter · Decision Letter 0]

4 Jun 2025

Dear Dr. Punsawad,

Thank you for submitting your manuscript to PLOS ONE. After careful consideration, we feel that it has merit but does not fully meet PLOS ONE’s publication criteria as it currently stands. Therefore, we invite you to submit a revised version of the manuscript that addresses the points raised during the review process.

**ACADEMIC EDITOR**

Dear Dr. Punsawad,

 Please ensure that all reviewer comments are carefully addressed in the revised version

We look forward to receiving your revised manuscript.

Kind regards,

José Luiz Fernandes Vieira

Academic Editor

PLOS ONE

Additional Editor Comments (if provided):

Reviewers' comments:

Reviewer's Responses to Questions

**Comments to the Author**

1. Is the manuscript technically sound, and do the data support the conclusions?

Reviewer #1: Yes

Reviewer #2: Yes

2. Has the statistical analysis been performed appropriately and rigorously?

Reviewer #1: Yes

Reviewer #2: Yes

3. Have the authors made all data underlying the findings in their manuscript fully available?

Reviewer #1: Yes

Reviewer #2: Yes

4. Is the manuscript presented in an intelligible fashion and written in standard English?

Reviewer #1: Yes

Reviewer #2: Yes

Reviewer #1: Major Comments for Revision

1. The dose of 600 mg/kg for FFR is relatively high. Please provide justification for selecting this dose based on prior toxicity studies or pharmacokinetic data. Also, clarify if dose-ranging studies were performed.

2. The combination of artesunate and FFR appears to yield additive or synergistic effects. Consider calculating a Combination Index (CI) or referring to the Chou–Talalay method to quantitatively assess drug interaction. This would strengthen the claim of synergy.

3. The extract is not chemically characterized beyond referencing past work. Please include or reference quantitative phytochemical profiling or HPLC fingerprints of the FFR extract used in this study for reproducibility.

4. While the use of ANOVA with Bonferroni post hoc is appropriate, effect sizes and confidence intervals for the key comparisons (e.g., parasitemia suppression, RMCBS, NOR) would provide more robust interpretations.

5. Cite the original sources or validated protocols for the Rapid Murine Coma and Behavior Scale and Novel Object Recognition test, especially if modifications were made.

6. The discussion would benefit from elaborating possible mechanistic pathways for FFR’s neuroprotection, e.g., antioxidant, anti-inflammatory, or blood-brain barrier stabilization effects. Consider integrating existing literature to support your hypotheses.

7. Minor grammatical errors and occasional awkward phrasing (e.g., “relieve dizziness,” “digestive enhancement”) appear throughout. A final native English editing pass is recommended for clarity and polish.

Minor Comments

- Clarify if investigators were blinded to treatment groups during scoring (RMCBS/NOR).

- Provide clearer captions for figures—indicating statistical significance, sample size, and scale bars.

Reviewer #2: General Evaluation:

This manuscript investigates the antimalarial and neuroprotective effects of the five-flower remedy (FFR), a traditional Thai herbal formulation, in a mouse model of cerebral malaria (CM). The authors provide compelling evidence that FFR, especially in combination with artesunate, can significantly reduce parasitemia, improve neurological function, and reduce neuroinflammation and neuronal damage. The study is timely, methodologically sound, and presents novel findings that may contribute to the development of adjunctive therapies for cerebral malaria. The manuscript is well written but it needs clarification in some points before being further considered for publication.

Recommendation: Minor Revisions

Major Strengths:

• The study addresses an important gap in cerebral malaria therapy, focusing on neuroprotective outcomes.

• The experimental design is well-structured, with appropriate control and treatment groups.

• The use of behavioral (RMCBS, NOR), histological, and molecular endpoints provides robust support for the findings.

• Results are clearly presented and discussed in a logical and contextualized manner.

Specific Comments:

Introduction

1. Consider summarizing FFR’s pharmacological background more concisely.

2. Clarify the novelty of the study—emphasize that this is the first in vivo assessment of FFR in CM.

Materials and Methods

1. At Plant preparation and extraction and Dosing of CM model, please explain the rationale behind the choice of 600 mg/kg FFR dose.

2. Clarify whether randomization and blinding were applied during group assignment and outcome assessment.

3. At the part of Dosing and grouping of CM model, why did the researcher start treatment on day 6-12, while behavioral changes were monitored on day 4?

Results

1. Should include Ct or melt curve data for qPCR in supplementary files to support gene expression results.

2. Please improve image resolution for histology figures and change the color of the arrowhead at Fig 6.

Discussion

1. Consider discussing the possibility of drug-herb interactions with artesunate.

2. Avoid redundancy—some background material from the introduction is repeated.

3. Recommend mentioning study limitations and suggesting future directions (e.g., mechanistic studies, dose-response, clinical evaluation).

Conclusion

- Strong conclusion supported by data.

Final Verdict: Minor Revisions

I recommend acceptance after minor revisions addressing methodological clarification and figure enhancement.

**Do you want your identity to be public for this peer review?** For information about this choice, including consent withdrawal, please see our Privacy Policy

Reviewer #1: No

Reviewer #2: No

---

## [Author Response · Author response to Decision Letter 1]

30 Jul 2025

A point-by-point response to the reviewer’s comments

Reviewer #1: Major Comments for Revision

1. The dose of 600 mg/kg for FFR is relatively high. Please provide justification for selecting this dose based on prior toxicity studies or pharmacokinetic data. Also, clarify if dose-ranging studies were performed.

Response:

We thank the reviewer for raising this important point regarding dose selection. The choice of 600 mg/kg for FFR administration in the cerebral malaria (CM) model was based on both efficacy and safety considerations derived from our preliminary studies. Specifically, we conducted a dose-ranging evaluation using the standard 4-day suppressive test in a Plasmodium berghei-infected mouse model. In this assay, three different doses of FFR (200, 400, and 600 mg/kg) were administered orally. The 600 mg/kg dose consistently demonstrated the most pronounced antiplasmodial activity, with a significantly higher percentage suppression of parasitemia compared to the lower doses (unpublished data). Moreover, to ensure that the selected dose would not pose safety concerns, we also performed an acute toxicity study in mice. Animals were administered a single oral dose of 2,000 mg/kg of FFR, and no mortality or behavioral abnormalities were observed throughout a 14-day observation period. Histopathological examination of the liver and kidney tissues using hematoxylin and eosin (H&E) staining revealed no pathological alterations. In addition, serum biochemical markers of liver and renal function (e.g., ALT, AST, ALP, creatinine, BUN) remained within normal limits, indicating no organ toxicity at this high dose (unpublished data). Together, these findings provided a rational basis for selecting the 600 mg/kg dose for the CM model, as it represents a therapeutically effective dose that remains well below the threshold for acute toxicity. While formal pharmacokinetic studies are ongoing, the absence of observable adverse effects at >3-fold higher dose (2,000 mg/kg) supports the safety margin of the selected treatment concentration. Moreover, we would like to highlight that previous in vivo studies investigating herbal formulations in cerebral malaria models have also employed similarly high doses, particularly in the range of 300-1,000 mg/kg, to ensure sufficient bioavailability at the site of infection and central nervous system involvement (1, 2). Our dosing choice is thus consistent with established practice in related experimental models.

References

1. Alharbi A, Albasyouni S, Al-Shaebi E, Al Quraishy S, Abdel-Gaber R. Neuroprotective and antimalarial effects of Juglans regia leaf extracts in a murine model of cerebral malaria. Frontiers in Veterinary Science. 2025;Volume 12 - 2025.

2. Bedri S, Khalil EA, Khalid SA, Alzohairy MA, Mohieldein A, Aldebasi YH, et al. Azadirachta indica ethanolic extract protects neurons from apoptosis and mitigates brain swelling in experimental cerebral malaria. Malaria journal. 2013;12:1-9.

2. The combination of artesunate and FFR appears to yield additive or synergistic effects. Consider calculating a Combination Index (CI) or referring to the Chou–Talalay method to quantitatively assess drug interaction. This would strengthen the claim of synergy.

Response:

We appreciate the reviewer’s insightful suggestion regarding the assessment of potential synergistic interactions between artesunate and FFR through the calculation of a Combination Index (CI) based on the Chou–Talalay method. We fully agree that such an approach would provide a quantitative evaluation of the observed combination effects. However, we would like to note that in our current study, the 4-day suppressive test employed a single fixed dose of artesunate (6 mg/kg) in combination with FFR. Dose-varying experiments were not conducted, and the data required to determine the IC₅₀ values for artesunate and FFR (both individually and in combination) were not collected. Consequently, we acknowledge that the current dataset does not allow for accurate CI calculation using the Chou–Talalay approach. Nonetheless, we appreciate the reviewer’s recommendation and recognize its importance. In future studies, we intend to design experiments that include dose-response curves for each compound and their combinations to evaluate drug interactions using CI values and isobologram analysis.

3. The extract is not chemically characterized beyond referencing past work. Please include or reference quantitative phytochemical profiling or HPLC fingerprints of the FFR extract used in this study for reproducibility.

Response:

Thank you very much for your valuable suggestion regarding the chemical characterization of the FFR extract. We would like to clarify that previous studies on FFR extract have focused primarily on preliminary phytochemical screening and traditional therapeutic applications. In response, we have now included the phytochemical profiling data of the FFR extract to ensure reproducibility and transparency of the experimental materials. These data are presented in the revised manuscript [please refer to page 28, lines 646-648].

4. While the use of ANOVA with Bonferroni post hoc is appropriate, effect sizes and confidence intervals for the key comparisons (e.g., parasitemia suppression, RMCBS, NOR) would provide more robust interpretations.

Response:

We sincerely thank the reviewer for the valuable suggestion regarding the inclusion of effect sizes and confidence intervals to support key comparisons. We agree that this addition would enhance the interpretability and robustness of our statistical findings. Accordingly, we have revised the manuscript to include effect size measures and 95% confidence intervals for the main outcome variables. Please refer to page 17, lines 378–380 of the revised manuscript for the updated statistical presentation.

5. Cite the original sources or validated protocols for the Rapid Murine Coma and Behavior Scale and Novel Object Recognition test, especially if modifications were made.

Response:

We thank the reviewer for highlighting the importance of referencing validated protocols for behavioral assessments. In response, we have added the original references for both the Rapid Murine Coma and Behavior Scale (RMCBS)and the Novel Object Recognition (NOR) test in the revised manuscript. These citations correspond to the standard protocols upon which our methodology was based. The updated references and clarifications are intended to strengthen the methodological of our study and ensure alignment with established standards in behavioral neuroscience research. We kindly invite the reviewer to refer to the revised version of the manuscript for these updates. Please refer to page 13, line 289 for RMCBS and page 13, line 303 for NOR test.

6. The discussion would benefit from elaborating possible mechanistic pathways for FFR’s neuroprotection, e.g., antioxidant, anti-inflammatory, or blood-brain barrier stabilization effects. Consider integrating existing literature to support your hypotheses.

Response:

Thank you very much for your thoughtful and constructive comment. As suggested, we have expanded the Discussion section to elaborate on the possible mechanistic pathways underlying the neuroprotective effects of FFR. Specifically, we discuss three interrelated mechanisms: (1) antioxidant activity that mitigates oxidative neuronal damage, (2) anti-inflammatory effects that modulate neuroinflammatory responses, and (3) the potential stabilization of the blood–brain barrier (BBB) which may prevent pathological infiltration of immune cells into the central nervous system.

These mechanistic insights have been integrated into our revised discussion and are directly aligned with the hypothesis of our study—that FFR exerts neuroprotective effects in cerebral malaria not only through symptomatic relief but also via molecular and cellular mechanisms that attenuate neuropathology. Please refer to the page 35-36, lines 815–833.

7. Minor grammatical errors and occasional awkward phrasing (e.g., “relieve dizziness,” “digestive enhancement”) appear throughout. A final native English editing pass is recommended for clarity and polish.

Response:

Thank you for your suggestion. We have carefully revised the manuscript to correct minor grammatical errors and improve awkward phrasing throughout the text. A final language editing has been completed to enhance clarity and ensure the manuscript meets academic writing standards.

Minor Comments

- Clarify if investigators were blinded to treatment groups during scoring (RMCBS/NOR).

Response:

Thank you for your comment. In accordance with standard practices to reduce observer bias, the assessments of both the Rapid Murine Coma and Behavior Scale (RMCBS) and the Novel Object Recognition (NOR) test were conducted by an independent observer who was blinded to the treatment group. Specifically, during data collection and scoring, the investigator was not informed of the treatment conditions assigned to each animal. This blinding process was implemented to ensure objectivity and consistency in behavioral evaluations. This procedure has also been stated in the revised version of the manuscript, and it is consistent with our previous methodology, where results were analyzed by an observer who was unaware of the experimental groupings. This information has been explicitly stated in the "Materials and Methods" section (page 13, lines 293-296 and page 14, lines 315–318).

- Provide clearer captions for figures—indicating statistical significance, sample size, and scale bars.

Response:

Thank you for your suggestion. I have now revised the manuscript accordingly by including statistical significance, sample sizes, and scale bars in the figure captions.

Reviewer #2: General Evaluation:

This manuscript investigates the antimalarial and neuroprotective effects of the five-flower remedy (FFR), a traditional Thai herbal formulation, in a mouse model of cerebral malaria (CM). The authors provide compelling evidence that FFR, especially in combination with artesunate, can significantly reduce parasitemia, improve neurological function, and reduce neuroinflammation and neuronal damage. The study is timely, methodologically sound, and presents novel findings that may contribute to the development of adjunctive therapies for cerebral malaria. The manuscript is well written but it needs clarification in some points before being further considered for publication.

Recommendation: Minor Revisions

Major Strengths:

• The study addresses an important gap in cerebral malaria therapy, focusing on neuroprotective outcomes.

• The experimental design is well-structured, with appropriate control and treatment groups.

• The use of behavioral (RMCBS, NOR), histological, and molecular endpoints provides robust support for the findings.

• Results are clearly presented and discussed in a logical and contextualized manner.

Response:

We sincerely thank the reviewer for their thoughtful and encouraging feedback. We greatly appreciate your recognition of the relevance and significance of our study in addressing a critical gap in cerebral malaria therapy, particularly regarding its neuroprotective potential.

Your supportive comments are highly motivating and deeply appreciated.

Specific Comments:

Introduction

1. Consider summarizing FFR’s pharmacological background more concisely.

Response:

Thank you for your helpful suggestion. In response, we have revised the pharmacological background of the FFR to present the information more concisely. The updated version has been incorporated into the revised manuscript. Please refer to page 5-6, lines 99–127.

2. Clarify the novelty of the study—emphasize that this is the first in vivo assessment of FFR in CM.

Response:

Thank you very much for your insightful suggestion. We have now clarified the novelty of the present study in the revised manuscript by emphasizing that this is the first in vivo investigation of the effects of the FFR in a cerebral malaria model. Please refer to page 6, lines 133–139 of the revised manuscript.

Materials and Methods

1. At Plant preparation and extraction and Dosing of CM model, please explain the rationale behind the choice of 600 mg/kg FFR dose.

Response:

We appreciate your insightful question regarding the rationale for dose selection. The selection of 600 mg/kg as the treatment dose for FFR in the cerebral malaria model was based on preliminary dose-ranging experiments conducted using the standard 4-day suppressive test in Plasmodium berghei-infected mice. In this assay, FFR was administered orally at doses of 200, 400, and 600 mg/kg, and antiplasmodial efficacy was evaluated based on the percentage suppression of parasitemia. Among the tested doses, 600 mg/kg consistently produced the most pronounced suppression, indicating superior efficacy over the lower doses. Although these data are currently unpublished, they served as the primary rationale for selecting this dose for subsequent in vivo studies in the CM model (1-2). In addition, results from acute toxicity testing demonstrated that a single oral dose of FFR at 2,000 mg/kg did not cause mortality, behavioral abnormalities, or histopathological changes in vital organs. These findings confirmed that 600 mg/kg is well within the safe therapeutic range, providing further justification for its use in the present study.

References

1. Alharbi A, Albasyouni S, Al-Shaebi E, Al Quraishy S, Abdel-Gaber R. Neuroprotective and antimalarial effects of Juglans regia leaf extracts in a murine model of cerebral malaria. Frontiers in Veterinary Science. 2025;Volume 12 - 2025.

2. Bedri S, Khalil EA, Khalid SA, Alzohairy MA, Mohieldein A, Aldebasi YH, et al. Azadirachta indica ethanolic extract protects neurons from apoptosis and mitigates brain swelling in experimental cerebral malaria. Malaria journal. 2013;12:1-9.

2. Clarify whether randomization and blinding were applied during group assignment and outcome assessment.

Response:

Thank you for this valuable comment. In response, we have now included a detailed description of the randomization and blinding procedures in the revised manuscript. Specifically, animals were randomly assigned to treatment groups. In addition, both treatment administration and outcome assessments—including behavioral evaluations and histopathological analyses—were conducted under blinded conditions to minimize the risk of observer bias and enhance the scientific rigor of the study. These methodological have been explicitly stated in the revised “Materials and Methods” section to ensure transparency and reproducibility. Please refer to page 10, lines 227 of the revised manuscript.

3. At the part of Dosing and grouping of CM model, why did the researcher start treatment on day 6-12, while behavioral changes were monitored on day 4?

Response:

Thank you for your thoughtful question. Behavioral monitoring was initiated on day 4 post-infection to establish a baseline for neurological changes and to track the progression of cerebral malaria before treatment began. In murine models of CM, clinical signs—such as reduced locomotor activity, diminished responsiveness, and impaired exploratory behavior—typically emerge around days 6 to 7 post-infection (1). Early behavioral assessments are therefore essential for identifying the onset of disease and assessing the neurological impairment prior to therapeutic intervention. Importantly, a previous study demonstrated that mice with relatively high RMCBS scores (>15) on day 4 already exhibited intracerebral hemorrhages upon histopathological examination, even in the absence of overt clinical symptoms or detectable parasitemia. In contrast, mice with lower RMCBS scores showed widespread hemorrhages throughout multiple brain regions (2). These findings highlight the value of early neurological evaluation, even before clinical signs are apparent, and support the timeline used in our experimental design. In our study, RMCBS scores in the infected untreated group had already declined to approximately 12 by day 6, suggesting the emergence of significant neurological symptoms and the potential for underlying cerebral pathology. Based on this observation, treatment was initiated on day 6 to c

---

## [Decision Letter · Decision Letter 1]

8 Aug 2025

Antimalarial and neuroprotective effects of ethanolic extracts of the five-flower remedy in an experimental cerebral malaria model

PONE-D-25-19467R1

Dear Dr.Chuchard Punsawad

We’re pleased to inform you that your manuscript has been judged scientifically suitable for publication and will be formally accepted for publication once it meets all outstanding technical requirements.

Kind regards,

José Luiz Fernandes Vieira

Academic Editor

PLOS ONE

Reviewers' comments:

**Comments to the Author**

Reviewer #1: All comments have been addressed

Reviewer #2: All comments have been addressed

2. Is the manuscript technically sound, and do the data support the conclusions?

Reviewer #1: Yes

Reviewer #2: Yes

3. Has the statistical analysis been performed appropriately and rigorously?

Reviewer #1: Yes

Reviewer #2: Yes

4. Have the authors made all data underlying the findings in their manuscript fully available?

Reviewer #1: Yes

Reviewer #2: Yes

5. Is the manuscript presented in an intelligible fashion and written in standard English?

Reviewer #1: Yes

Reviewer #2: Yes

Reviewer #1: I would like to commend the authors for their thorough and careful revision of the manuscript titled “Antimalarial and neuroprotective effects of ethanolic extracts of the five-flower remedy in an experimental cerebral malaria model.” All reviewer comments, both major and minor, have been clearly addressed with appropriate revisions made throughout the manuscript. The authors have provided detailed justifications, included additional data and references where necessary, improved the clarity and scientific rigor of the methods and discussion, and ensured transparency in experimental design and reporting.

Reviewer #2: The manuscript is well-written and presents interesting findings that are relevant to the field. The study design and data are generally sound, and the results are clearly presented. Minor language polishing and a few clarifications in the methods section would further improve the manuscript. Overall, it is a valuable contribution and suitable for publication.

**Do you want your identity to be public for this peer review?** For information about this choice, including consent withdrawal, please see our Privacy Policy

Reviewer #1: **Yes: ** Assoc. Prof. Voravuth Somsak, MT, Ph.D., SFHEA

Reviewer #2: No

---

## [Editor Report · Acceptance letter]

PONE-D-25-19467R1

PLOS ONE

Dear Dr. Punsawad,

I'm pleased to inform you that your manuscript has been deemed suitable for publication in PLOS ONE. Congratulations! Your manuscript is now being handed over to our production team.

Kind regards,

on behalf of

Dr. José Luiz Fernandes Vieira

Academic Editor

PLOS ONE